# Small RNA GcvB Regulates Oxidative Stress Response of *Escherichia coli*

**DOI:** 10.3390/antiox10111774

**Published:** 2021-11-05

**Authors:** Xian Ju, Xingxing Fang, Yunzhu Xiao, Bingyu Li, Ruoping Shi, Chaoliang Wei, Conghui You

**Affiliations:** 1Shenzhen Key Laboratory of Microbial Genetic Engineering, College of Life Sciences and Oceanology, Shenzhen University, Shenzhen 518055, China; 2170257314@email.szu.edu.cn (X.J.); fang_xing_xing@163.com (X.F.); xiaoyunzhu8891@163.com (Y.X.); byli@szu.edu.cn (B.L.); srp642015142@163.com (R.S.); 2Health Science Center, Shenzhen University, Shenzhen 518055, China; weicl@szu.edu.cn

**Keywords:** *Escherichia coli*, small RNA, GcvB, oxidative stress, OxyR

## Abstract

Small non-translated regulatory RNAs control plenty of bacterial vital activities. The small RNA GcvB has been extensively studied, indicating the multifaceted roles of GcvB beyond amino acid metabolism. However, few reported GcvB-dependent regulation in minimal medium. Here, by applying a high-resolution RNA-seq assay, we compared the transcriptomes of a wild-type *Escherichia coli* K-12 strain and its *gcvB* deletion derivative grown in minimal medium and identified putative targets responding to GcvB, including *flu*, a determinant gene of auto-aggregation. The following molecular studies and the enhanced auto-aggregation ability of the *gcvB* knockout strain further substantiated the induced expression of these genes. Intriguingly, the reduced expression of OxyR (the oxidative stress regulator) in the *gcvB* knockout strain was identified to account for the increased expression of *flu*. Additionally, GcvB was characterized to up-regulate the expression of OxyR at the translational level. Accordingly, compared to the wild type, the GcvB deletion strain was more sensitive to oxidative stress and lost some its ability to eliminate endogenous reactive oxygen species. Taken together, we reveal that GcvB regulates oxidative stress response by up-regulating OxyR expression. Our findings provide an insight into the diversity of GcvB regulation and add an additional layer to the regulation of OxyR.

## 1. Introduction

The *Escherichia coli* chromosome encodes more than 80 small non-translated regulatory RNAs [1] that control plenty of bacterial vital activities, including biofilm formation [2], transcription termination [3], cell signaling [4] and cellular responses to growth conditions of various factors of oxidative stress, osmolarities, temperature or iron levels [5,6,7,8,9,10,11,12].

The small RNA GcvB has been extensively studied since its discovery two decades ago [5]. GcvB is approximately 200 nt in length and is highly conserved among Enterobacteriaceae [13,14]. GcvB expression is tightly related to glycine metabolism [15,16,17,18,19] and GcvA activates the transcription of GcvB [20] when glycine is abundant. Intriguingly, GcvB expression shows a strong growth condition-dependent feature. GcvB is abundant when cells are grown in rich medium during the exponential phase but is deficient when cells reach the stationary phase or are grown in minimal medium [5,13,15,21]. GcvB is depicted as a regulator of 1–2% of all the mRNAs in *Salmonella Typhimurium* and *E. coli* by using genome-wide experimental approaches including microarray [16] and RIL-seq [14], or by applying network [22] or in silico prediction [16,17]. Most of the known targets negatively regulated by GcvB are amino acid biosynthesis proteins and transporters of amino acids or peptides, as exampled by the two asparagine synthetases of AsnA and AsnB [21], the periplasmic transporter components of DppA and OppA [20], the serine/threonine transporter of SstT [23], and a certain amino acids transporter of CycA [15], suggesting the primary physiology role of GcvB could be to limit the uptake and biosynthesis of energy-expensive amino acid under nutrient-rich conditions [13,20]. In parallel, GcvB also negatively regulates the expression of several transcription factors including Lrp [24], a global regulator responding to the intracellular leucine level [25]; PhoP [26] in the PhoQP two-component system, which controls the expression of genes with functions in magnesium transport, acid resistance and lipopolysaccharide modification [27,28,29]; and CsgD [30], the master regulator of curli synthesis [31] Moreover, in a direct or an indirect mode, GcvB is positively involved in the acid stress response [32] and stress-induced DNA mutagenic break repair [33]. All this evidence indicates the relatively multifaceted roles of GcvB that are beyond amino acid metabolism.

In most cases, GcvB functions as a repressor. It blocks translational initiation by classically interfering with the 30S ribosome subunit binding to a ribosome-binding site or inducing an active mRNA decay via the recruitment of ribonuclease E after pairing to the target mRNA [21]. GcvB also works as an activator. GcvB maintains the stability of the *rbn* mRNA with the help of Hfq and protects it from RNase E cleavage [34]. GcvB up-regulates sigma factors RpoS [32] and RpoE [33], although the precise nature of the interactions between GcvB and these sigma factors is to be determined.

Therefore, there is diversity underlining both of the physiological roles and the molecular mechanisms of GcvB, which is worthy of further extensive exploration. Since people were unable to identify the GcvB-dependent regulation in minimal medium using traditional methods as promoter-*lacZ* fusions [15,20,21,23], possibly owing to the low expression of GcvB [21], in the present work, by applying the high resolution RNA-seq assay [35] that enabled us to largely expand the members in the regulons of several global regulators [36,37], we identified several novel targets putatively responding to GcvB in an *E. coli* K-12 strain grown in minimal medium. The subsequent molecular and physiological studies enabled us to discover that GcvB controlled the oxidative stress response of *E. coli* via the up-regulation of OxyR, the master regulator of antioxidant genes [38,39]. Thus, our findings not only provide insight into the diversity of GcvB regulation but also add an additional layer to the regulation of OxyR.

## 2. Materials and Methods

### 2.1. Construction of Bacterial Strains and Plasmids

All strains used in this work were derived from the wild-type *E. coli* K-12 strain NCM3722 [40] (GenBank acc. n: CP011496.1) and listed in Appendix A. Its derivatives of CY325 (Δ*gcvB*), CY1057 (Δ*oxyR*::*cat*) and CY1042 (Δ*gcvB oxyR*::*cat*) were constructed using the λ-Red system and the antibiotics markers were removed as described previously when necessary [41]. In aggregation-related experiments, since NCM3722 was unable to form flagella due to the defect of the flagella-structural protein FliC [42], the mutated *fliC* in NCM3722 was repaired to the wild-type *fliC* in MG1655 via P1 transduction to regain motility, resulting in the strain CY713. Then, its derivatives of the *gcvB* knockout strain (CY1027) and the *oxyR* knockout strain (CY1038) were, respectively, constructed by the λ-Red system [41]. In order to detect OxyR using Western blot, the myc tag was, respectively, inserted right before the stop codon of the *oxyR* gene in the *gcvB* wild-type strain NCM3722 and in the *gcvB* knockout strain CY325 by the λ-Red system [41], resulting in strains CY454 and CY455. To construct the *oxyR* promoter with *lacZ* fusion strains, at a first step, the *oxyR* promoter region (−266 bp to +99 bp relative to *oxyR* translational start point) was amplified from the genomic DNA of NCM3722 and was then inserted into the kpn I and Hind III sites of plasmid pCY161 (modified from pUC19 by adding an FRT-neo-FRT via EcoR I site), yielding plasmid pCY161-oxyR. Subsequently, the DNA regions of P1*oxyR* (−266 bp to +99 bp relative to *oxyR* translational start point) or P2*oxyR* (−266 bp to +45 bp relative to *oxyR* translational start point) together with the *neo* gene, was PCR amplified from the pCY161-oxyR plasmid and, respectively, integrated into the chromosome of the *gcvB* wild-type strain CY1057 and the *gcvB* knockout strain CY1042 to replace part of *lacI* and the entire *lacZ* promoter with the 5′UTR of *lacZ* (from +134 bp after *lacI* translational start codon to *lacZ* translational start codon) using the λ-Red system [41], resulting in the translational fusion strains CY1037 (P1*oxyR* in the *gcvB* wild-type background), CY1041 (P1*oxyR* in the *gcvB* knockout background), CY1059 (P2*oxyR* in the *gcvB* wild-type background) and CY1060 (P2*oxyR* in the *gcvB* knockout background), individually. P3*oxyR* (−266 bp to +0 bp relative to *oxyR* translational start point) was also PCR amplified from pCY161-*oxyR* and, respectively, integrated into the chromosome of CY1057 and CY1042 to replace part of *lacI* and the promoter of *lacZ* (from +134 bp after *lacI* translational start codon to *lacZ* transcriptional start site) using the λ-Red system [41], resulting in the transcriptional fusion strains of CY1047 (P3*oxyR* in the *gcvB* wild-type background) and CY1052 (P3*oxyR* in the *gcvB* knockout background). Note that we deleted the gene of *oxyR* in all of the *lacZ* fusion strains because *oxyR* expression was repressed by OxyR itself [43] and the LacZ activities of the *oxyR* promoters with the *lacZ* fusions were hard to detect in the presence of the native *oxyR*.

### 2.2. Growth of Cell Cultures

Batch cultures were grown in either the N^−^C^−^ minimal medium [40] supplied with 0.4% glucose and 20 mM NH_4_Cl or the LB rich medium. Batch culture growth in minimal medium was carried out in three steps in a 37 °C water bath shaker as described previously [40]. Batch culture growth in LB rich medium was carried out in two steps in a 37 °C water bath shaker: (1) Strains were firstly grown in LB overnight; (2) Then, they were diluted at a ratio of 1:100 for growth in LB of the experimental study. Cell samples were collected mid-exponential phase with an OD_600_ ~0.4 for the following transcriptome study and RT-qPCR assay.

### 2.3. RNA-Seq Assay and Analysis of Transcriptome Data

Total RNA extraction and RNA-seq assay were performed as described previously [36]. Two independent total RNA extractions and transcriptome analyses by RNA-seq were performed for each NCM3722 and CY325 grown in glucose minimal medium. The DEGs between the two transcriptomes were characterized by edgeR [*q*-value (FDR) < 0.01 and |log2FC| > 1]. FPKM (Fragments Per Kilobase of transcript per Million mapped reads) [44] was used to calculate the expression of each gene.

### 2.4. Quantitative Reverse Transcription PCR (RT-qPCR) Assay

The total RNA of NCM3722 and CY325 were extracted, and cDNA templates were synthesized following the instruction of PrimeScript RT reagent Kit with gDNA Eraser (Takara, Beijing, China). RT-qPCR assays were performed on the qTOWER3 real-time PCR thermo-cycler (Analytik Jena, Jena, Germany). Primers for RT-qPCR were designed using Primer3 (v. 0.4.0) and listed in Appendix A. The gene of *recA* was used as a reference control. The reaction mixture (20 μL) consisted of 10 μL of TB Green Premix Ex Taq II (Tli RNaseH Plus) (Takara, Beijing, China), 2 μL of template (10-fold diluted cDNA), 7 μL of H_2_O and 0.5 μL (10 μM) of each primer. No template controls (NTC) were run in parallel, and all reactions were performed in triplicate. The relative quantification (RQ) was analyzed using the ΔΔCt quantification method by qPCRsoft v3.2, where CT refers to the cycle threshold.

### 2.5. Aggregation Assay with Time-Lapse Microscopy

*E. coli* cells were grown in LB to log phase with an OD_600_ ~ 0.4. Cells were then collected by centrifugation (3 min, 6000 rpm) and resuspended in LB to a final OD_600_ of 0.1. Then, 2-milliliter cell suspensions were loaded into a 24-well polystyrene plate and cell clumps was also observed at 37 °C, using a phase-contrast microscopy (Nikon Ti2, Tokyo, Japan) with 20× objective lens and PH-20× -G eyepiece, which had a total 400× magnification effect.

### 2.6. Western Blot Assay

Strains were grown in LB or glucose minimal medium to log phase with an OD_600_ ~ 0.4. Cells were centrifuged at 6000 rpm for 3 min and resuspended with 1 × PBS. Then, 0.025 OD_600_ or 0.05 OD_600_ cells were mixed with M5 SDS-PAGE loading buffer (Mei5 biotechnology, Beijing, China) and then boiled for 10 min. All the processed samples were added into SDS-PAGE gel and run at 120 V for approximately 120 min. After running the gel, the protein on the gel was transferred to the PVDF (polyvinylidene fluoride) membrane (GE health care life science, Beijing, China) at 110 V for about 40 min. Additionally, then the PVDF membrane was supposed to be immersed in 5% skimmed milk powder prepared from TBST (20 mM Tris, 150 mM NaCl and 0.1% Tween-20) and shaken at room temperature for 3 h. In this experiment, the selected internal reference protein was RpoB. Before incubating with different antibodies, the membrane was cut into two pieces according to the size of the protein and washed with TBST. The primary antibody (Monoclonal anti-RpoB antibody and Anti-Myc tag antibody) was added at a ratio of 1:2000 and was incubated with the membrane at room temperature for 2 h on a shaker. The secondary antibody (Goat anti-mouse lgG-HRP conjugated secondary antibody and Goat anti-rabbit lgG-HRP conjugated secondary antibody) was added at a ratio of 1:4000 and incubated with membranes at room temperature for 1.5 h on a shaker. SuperSignal West Pico PLUS (Thermo, Rockford, IL, USA) was applied for the exact purpose of imaging exposure. The image J was used for quantification analysis.

### 2.7. β-Galactosidase Activity Assay

Overnight culture of *E. coli* cells in LB were diluted at a ratio of 1:100 and grown in the same medium. At various OD_600_ (~0.2–~0.4) following the growth, three to five cell samples (1 mL) were quickly frozen in liquid nitrogen. Then, 0.025–0.5-milliliter (V) thawed samples were added into each tube and mixed with 0.975–0.5 mL of Z-buffer (60 mM Na_2_HPO_4_, 40 mM NaH_2_PO_4_, 10 mM KCL and 1 mM MgSO_4_), containing BME (β-Mercaptoethanol) to be a total reaction volume of 1 mL. One blank reaction with solely 1 mL of Z-buffer was set. Additionally, 25 μL of 0.1% SDS and 50 μL of chloroform were added into the commixture. After incubating these tubes in the 37 °C water bath for 5 min, at time T_0_, 200 μL of 4 mg/mL ONPG (o-Nitrophenyl β-D-galactopyranoside) were added into each mixture and incubated in the water bath at 37 °C. When the color of the reaction turned yellow at time T_1_, we stopped the reaction by adding 1 M Na_2_CO_3_. The reaction time T = T_1_ − T_0_ (min). Finally, 1 mL of reaction sample was centrifuged at 13,000 rpm for 5 min and the OD_420_ and OD_550_ were measured. The β-galactosidase activity in miller unit (U/mL/OD_600_) was calculated as 1000 × (OD_420_ − 1.75 × OD_550_)/V/T.

### 2.8. Assay of Bacterial Resistance to H_2_O_2_ Treatment

Overnight cultures of the strains in LB were diluted at a ratio of 1:100 in the same medium and cultured to an exponential phase. Subsequently, the cultures were individually diluted to an OD_600_ of 0.05 and treated with 3 mM H_2_O_2_. Then, the OD_600_ of each culture was measured. For survival assay, bacterial cells grown in LB from early exponential phase (OD_600_ of ~0.4) were exposed to 20 mM H_2_O_2_ for 10 min and plated on LB plates by series dilution. The survival rate was calculated by colony counting.

### 2.9. Microscopic Detection of Reactive Oxygen Species (ROS)

NCM3722 and CY325 were incubated overnight in 2 mL of LB, after which they were diluted at a ratio of 1:100 into LB and grown to log phase with an OD_600_ of ~0.4. Additionally, 1 mL of cell culture of each strain was harvested by centrifugation and resuspended into 1 mL of sterile PBS. Then, 10^8^ cells/mL were collected and incubated with 150 μM DHE and H_2_DCFDA for 90 min in darkness with rotation at 37 °C. After staining, the bacteria were centrifuged at 6000 rpm for 3 min at room temperature and washed with the same volume of warm sterile PBS. Finally, a cell density of approximately 2 × 10^7^ cells/mL was adjusted for microscopic observation. Cells were observed at 37 °C with a phase-contrast microscopy (Nikon Ti2, Tokyo, Japan). A 20×objective lens and PH-20×-G eyepiece were applied for observing phase-contrast cells. A 450-nanometer laser was used for excitation and a band-pass filter of 610 nm was used to collect the red fluorescence. A 480-nanometer laser was used for excitation and a band-pass filter of 535 nm was used to collect the green fluorescence. Fluorescence intensity of each cell was quantified using the software of Nikon NIS-Elements AR.

## 3. Results

### 3.1. The Expression of Flu Was Highly Induced in the gcvB Deletion Strain

To characterize the putative target genes controlled by the small RNA GcvB (GenBank acc. n: CP011496.1, region 4,184,713–4,184,917), we compared the transcriptomes of the wild-type *E. coli* K-12 strain NCM3722 and its *gcvB* deletion derivative CY325 grown in glucose minimal medium by high resolution RNA-seq [35]. We first found that both the *gcvB* deletion and wild-type strains showed similar doubling times in the glucose minimal medium, indicative of a relatively limited role of GcvB regulation in this condition (Appendix A). In total, five genes were retained as significant with a two-fold cutoff (with an FDR smaller than 0.01) (Figure 1A, Appendix A) in the transcriptome study. Compared to their expression in NCM3722, the genes of *flu* (GenBank acc. n: CP011496.1, region 3,331,422–3,334,541) and *yeeR* (GenBank acc. n: CP011496.1, region 3,334,662–3,336,223) exhibited highly increased expression in CY325, respectively, showing a fold change of 63.6 and 4.6, whereas the remaining three genes showed an approximately two-fold decreased expression in CY325. Since *flu* and *yeeR* showed the biggest fold of expression changes*,* we further tested their expression in the two strains using the RT-qPCR assay. Consistent with the RNA-seq data, the mRNA levels of *flu* and *yeeR* in CY325 were significantly higher than that in the wild type when cells were grown in glucose minimal medium (Figure 1B,C). Similar results were also observed when cells were grown in LB rich medium (Figure 1D,E). Thus, *flu* and *yeeR* could be negatively regulated by GcvB in a direct or an indirect mode.

In the genome of NCM3722, the gene of *flu* localized right upstream of the gene of *yeeR* in the same DNA strand. The gene of *yeeR* is a pseudogene and the gene of *flu* encodes antigen 43 (Ag43). Ag43 is an abundant outer membrane protein that belongs to the auto-transporter family and is a major determinant of auto-aggregation in *E. coli* [45,46]. Thus, we studied the auto-aggregation of both the *gcvB* knockout and wild-type strains. As reported, bacterial motility and chemotaxis were required for the Ag43-dependent aggregation of *E. coli* [46]. However, the two strains applied in the transcriptome study were non-motile [47] because the flagellar filament structural protein [48] encoded by *fliC* (GenBank acc. n: CP011496.1, region 3,261,993–3,263,489) carries a lone mis-sense mutation (N87K) [42]. We complemented the mutated *fliC* of NCM3722 to the wild-type *fliC* to obtain the motile strain CY713, and the gene of *gcvB* was deleted in CY713 to obtain the *gcvB* knockout strain CY1027. Consequently, the auto-aggregation ability of CY713 and CY1027 was then compared using microscopy. Indeed, we observed that the *gcvB* knockout strain CY1027 aggregated more efficiently than the *gcvB* wild-type strain CY713 (Figure 2), supporting the highly induced *flu* expression in the *gcvB* knockout strain.

### 3.2. OxyR Showed Decreased Expression in the gcvB Deletion Strain

As reported, the transcription of *flu* was repressed by OxyR [49]. Accordingly, we identified the highly induced expression of *flu* in the *oxyR* (GenBank acc. n: CP011496.1, region 736,030–736,947) deletion strain (Appendix A). In parallel, the *oxyR* deletion strain was observed to exhibit the high auto-aggregation efficiency (Figure 2) indicative of the highly expressed *flu*. Moreover, we monitored the expression of *yeeR* in the *oxyR* deleted background and found that *yeeR* showed a similar induced expression as *flu* (Appendix A). Given their congruent responses and their concatenated locations in the genome, it was very likely that *flu* and *yeeR* formed an operon and both were under the control of OxyR.

Thus, the protein levels of OxyR in the *gcvB* knockout and wild-type strains grown in the LB rich medium or the glucose minimal medium were determined using Western blot. We identified that the protein amount of OxyR in CY455 (Δ*gcvB*) was approximately half that in the *gcvB* wild-type strain CY454 in both conditions (Figure 3). As a result, the decreased expression of the OxyR repressor in the *gcvB* knockout strain could induce the high expression of *flu*. 

### 3.3. The Small RNA GcvB Enhanced the Expression of OxyR at the Translational Level

We next explored how GcvB stimulated the expression of OxyR. The mRNA level of *oxyR* did not show significant changes in the two transcriptomes of the *gcvB* wild-type and knockout strains (Appendix A) and this finding was further demonstrated using the RT-qPCR assay (Appendix A). Moreover, we made an *oxyR* promoter with *lacZ* transcriptional fusion (Appendix A) in both the *gcvB* wild-type and knockout strains and observed that the β-galactosidase activity showed no significant changes in the two backgrounds (Appendix A). As a result, it was most likely that the regulation of GcvB on OxyR existed at the post-transcriptional level. To substantiate this hypothesis, we constructed the *oxyR* promoter with *lacZ* translational fusions in both the *gcvB* wild-type and knockout strains. We made two fusion constructions, with P1 and P2, respectively, carrying 99 and 45 nt after the translational start codon of *oxyR* (Figure 4A). Supporting the Western blot result (Figure 3), both translational fusions showed significantly decreased β-galactosidase activity in the *gcvB* knockout strain when being compared to that in the *gcvB* wild-type strain (Figure 4B,C), indicating GcvB activated the expression of OxyR at the translational level.

### 3.4. The gcvB Deletion Strain Was Sensitive to Oxidative Stress

Since OxyR is the transcriptional regulator that induces the expression of antioxidant genes in response to oxidative stress [38,39], the reduced expression of OxyR when *gcvB* was deleted suggested that the *gcvB* deletion strain could be sensitive to oxidative stress. Therefore, we tested the response of the *gcvB* wild-type and knockout strains to oxidative stress. Indeed, when additional H_2_O_2_ was supplied to the growth medium, the *gcvB* mutant grew more slowly and showed less resistance to H_2_O_2_ than the *gcvB* wild-type strain (Figure 5A). Moreover, we also tested the response of the *oxyR* deletion strain to H_2_O_2_ treatment. As expected, the *oxyR* deletion strain was sensitive to the oxidative stress. In the same line, both the *gcvB* and the *oxyR* mutants showed significantly lower survival rates than the wild type in the survival assay (Figure 5B).

The endogenous ROS were generated as byproducts of aerobic respiration [50]. Given the decreased expression of OxyR in the *gcvB* mutant, the *gcvB* mutant could accumulate more endogenous ROS. To verify this point, using fluorescence microscopy, we monitored the intracellular amount of ROS that remained in the *gcvB* wild-type and knockout strains by employing two kinds of ROS indicators, DHE (dihydroethidium) and H_2_DCFDA (2′,7′-dichloro-dihydro-fluorescein diacetate), both of which are membrane-permeant dyes that can be oxidized to show red and green fluorescence, respectively, and have been widely used as fluorescence probes for the detection of superoxide and peroxide due to their specificity to these radicals [51]. The phase contrast and fluorescence micro-photographs of the two strains stained with DHE and H_2_DCFDA were exhibited (Figure 6A and Figure 7A). Compared to the *gcvB* wild-type cells, larger numbers of *gcvB* knockout cells showed stronger fluorescence using both indicators indicative of higher levels of ROS remaining in the *gcvB* knockout cells. The mean fluorescence intensity of CY325 cells, respectively, stained with DHE or H_2_DCFDA was approximately two- to six-fold stronger than that of NCM3722 cells (Figure 6B and Figure 7B). These observations suggested that, compared to the *gcvB* wild-type cells, the *gcvB* knockout cells could lose some of their ability to eliminate endogenous ROS where the expression of OxyR was reduced.

## 4. Discussion

In this study, we performed comparative transcriptomes combined with molecular and physiological studies of the GcvB knockout and wild-type strains to characterize the putative target genes controlled by the small RNA GcvB. In this process, we were able to identify GcvB as a positive regulator controlling the oxidative stress response of *E. coli*.

GcvB-dependent regulation on its known targets in glucose minimal medium was almost undetectable using a traditional method [15], possibly because of the low expression of GcvB in this medium when being compared to its expression in rich medium [21]. In contrast, here, we characterized novel targets putatively controlled by GcvB in the minimal medium by applying the high-resolution RNA-seq assay that had helped us to successfully expand the members in the regulons of several global regulators [36,37]. Note that we also did not find the reported targets of GcvB in this medium, which is inconsistent with previous results.

We firstly found that the expression of *flu* and *yeeR* was highly induced when GcvB was deleted by the transcriptome study (Figure 1A), indicating that GcvB could negatively regulate their expression in a direct or indirect mode. Our following RT-qPCR studies (Figure 1B–E) and the enhanced *flu*-dependent auto-aggregation ability [45] of the *gcvB* knockout strain (Figure 2) further substantiated the induced expression of these genes.

We next explored the putative mechanisms of the small RNA GcvB repressed expression of *flu* and *yeeR*. It is well known that small RNA can regulate gene expression by controlling mRNA stability [52]. Consequently, GcvB could control the expression of *flu* and *yeeR* by regulating their mRNA stability as in the cases of its reported targets [21]. However, we found that *flu* and *yeeR* could form an operon since *flu* and *yeeR* were concatenated genes in the genome and they showed congruent expression changes in the conditions we studied (Figure 1 and Appendix A). To our knowledge, it was rare that a small RNA simultaneously regulated the expression of individual genes in one operon. As a result, it was more likely that GcvB controlled the expression of *flu* and *yeeR* in an indirect mode. Additionally, a mediator would coordinate the regulation of GcvB on *flu* and *yeeR*. Since OxyR, the oxidative stress regulator [38,39], is the reported transcriptional repressor of *flu* [49], this motivated us to study OxyR. Similar to *flu*, *yeeR* also responded to OxyR (Appendix A). Intriguingly, we identified that GcvB positively controlled the expression of OxyR (Figure 3), and this regulation existed at the translational level (Figure 4). Thus, OxyR could be the mediator and its reduced expression in the *gcvB* knockout strain could induce the high expression of *flu* and *yeeR*. However, further detailed studies will be required to investigate the nature of the interaction between GcvB and OxyR at the molecular level. 

Finally, our physiological studies supported the positive regulation of GcvB on OxyR since the GcvB deletion strain was not only more sensitive to external oxidative stress (Figure 5) but also lost some of its ability to eliminate endogenous ROS when being compared to the wild type (Figure 6 and Figure 7). Thus, the reduced expression of OxyR could account for these phenomena observed. However, except *flu*, we did not identify the other reported targets of OxyR [49] that showed significant expression changes between the GcvB wild-type and knockout strains using the RNA-seq assay, possibly because their expression changes could be small, thus, were below the detection limit of this method. Consequently, the OxyR regulon gene of *flu* could be the one that was most sensitive to the titration of OxyR. Moreover, it was proposed that the sensibility of bacteria to antibiotics was correlated to the endogenous production of ROS [53]. Therefore, the enhanced ROS levels in the *gcvB* knockout strain could make this strain more sensitive to antibiotics treatment and GcvB could accordingly work as a putative antimicrobial target.

## 5. Conclusions

By integrating a global assay with molecular and physiological studies, we reveal a novel function of the small RNA GcvB. GcvB enhances the survival ability of *E. coli* when experiencing oxidative stress by the up-regulation of OxyR expression. Our findings provide insight into the control of the oxidative stress response of *E. coli* by GcvB and add an additional layer to the regulation of OxyR. 

## Figures and Tables

**Figure 1 antioxidants-10-01774-f001:**
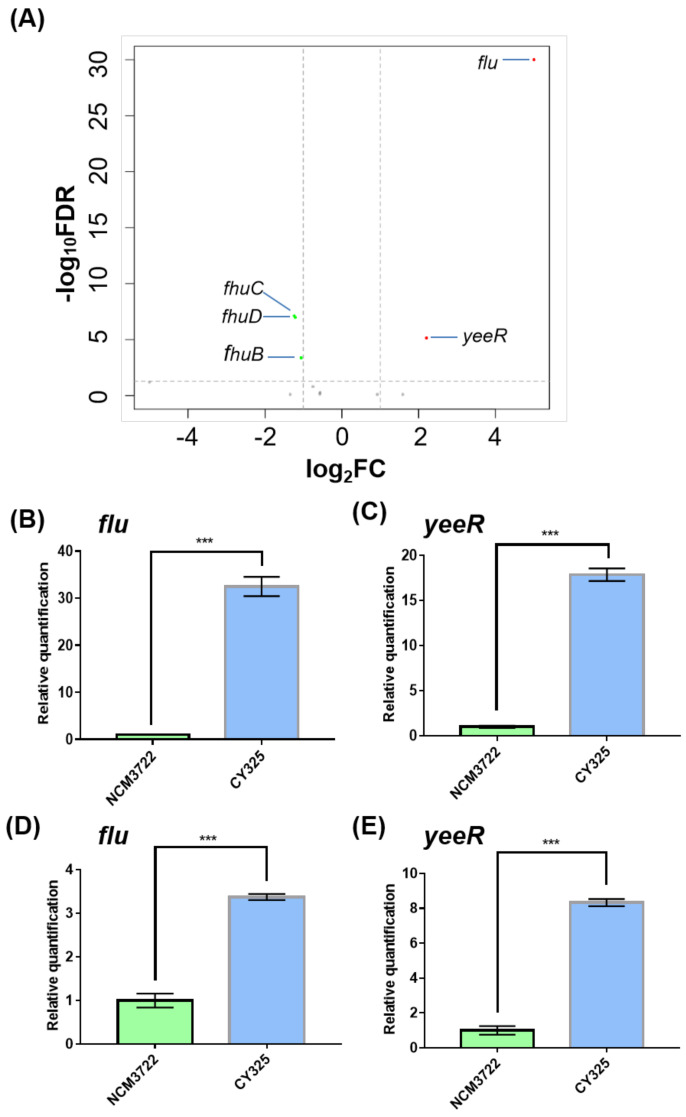
Genes responding to GcvB. (**A**) Volcano map comparing the expression of genes in the *gcvB* knockout strain CY325 versus that in the wild-type strain NCM3722. The horizontal dashed line points to *q*-value (FDR) = 0.01 on *y*-axis and the vertical dashed lines point to 2-fold cutoff of the expression on *x*-axis. Red dots indicate up-expressed genes; green dots indicate down-expressed genes; grey dots indicate genes showed no significant changes. Note that the data points of the remaining genomic genes carrying a *q*-value (FDR) = 1 with a corresponding −log_10_FDR = 0 were not shown. (**B**,**E**) Gene expression of *flu* and *yeeR* detected using RT-qPCR in the *gcvB* knockout strain CY325 and in the wild-type strain NCM3722 grown in glucose minimal medium (**B**,**C**) or in LB rich medium (**D**,**E**). The mRNA level of each gene in NCM3722 was normalized to 1 and that in CY325 was determined relative to this value. The relative expression was shown as the average ± S.D. of three independent experiments. *** *p* < 0.001 using Student’s *t*-test.

**Figure 2 antioxidants-10-01774-f002:**
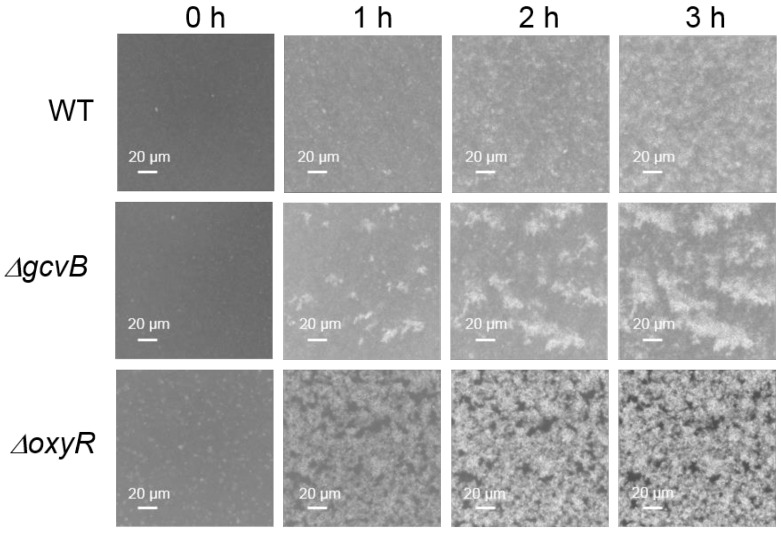
Aggregation assays of three *E. coli* strains. The aggregation of CY713 (WT), CY1027 (*gcvB*::cm) and CY1038 (*oxyR*::cm) was detected at 37 °C into a 24-well polystyrene plate using microscopy as described in methods. Bacterial aggregation was monitored at various time points (0–3 h). The formation of the white cluster indicates aggregation of cells. One representative result of two independent observations is shown.

**Figure 3 antioxidants-10-01774-f003:**
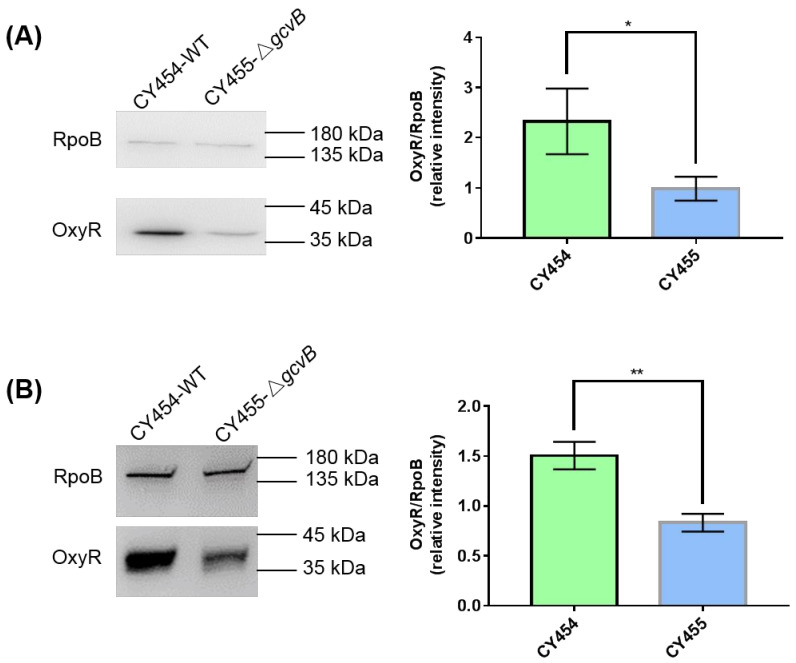
The protein levels of OxyR in the *gcvB* knockout and wild-type strains. The protein levels of OxyR in the *gcvB* wild-type strain CY454 and in the *gcvB* knockout strain CY455 cultured in LB rich medium (**A**) or in glucose minimal medium (**B**) were tested using Western blot. RpoB was applied as a loading control. The 0.025 OD_600_ (**A**) or 0.05 OD_600_ (**B**) cells were loaded. One representative result of six repeats in two independent Western blots is shown on the left. The relative protein level of three repeats in one Western blot is shown as the average ± S.D. on the right. The protein level of OxyR in CY455 was normalized to 1 and that in CY454 was determined relative to this value. ** *p* < 0.01, * *p* < 0.05 using Student’s *t*-test.

**Figure 4 antioxidants-10-01774-f004:**
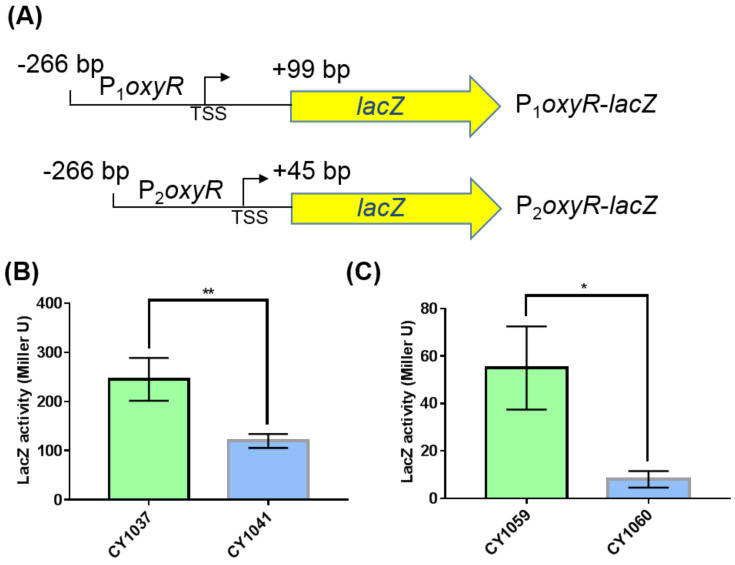
β-galactosidase activities of the *oxyR* promoter with *lacZ* translational fusions in the *gcvB* knockout and wild-type strains. (**A**) Composition diagrams of the promoters of *oxyR* with *lacZ* translational fusions. P_1_*oxyR* covers the DNA region from −266 bp to +99 bp relative to *oxyR* translational start point and P_2_*oxyR* covers the DNA region from −266 bp to +45 bp relative to *oxyR* translational start point. Yellow arrow indicates the entire coding region of *lacZ*. TSS indicates the transcriptional start site of *oxyR*. (**B**) β-galactosidase activities of P_1_o*xyR*-*lacZ* fusion in the *gcvB* wild-type (CY1037) and knockout strains (CY1041). (**C**) β-galactosidase activities of P_2_*oxyR*-*lacZ* fusion in the *gcvB* wild-type (CY1059) and knockout strains (CY1060). The LacZ activity was shown as the average ± S.D. of three independent experiments. ** *p* < 0.01, * *p* < 0.05 using Student’s *t*-test.

**Figure 5 antioxidants-10-01774-f005:**
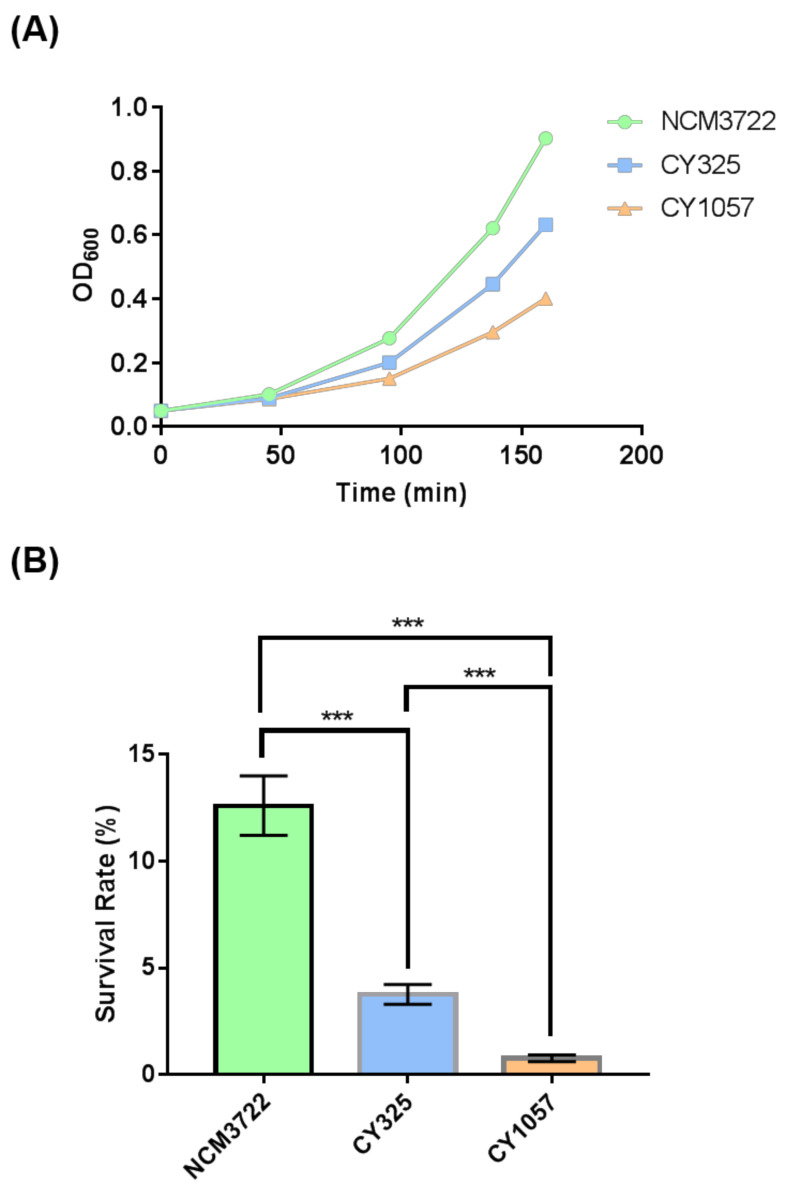
The sensitivity of the *gcvB* deletion strain to oxidative stress. (**A**) The growth of strains with H_2_O_2_ supplied to the medium. The growth curve of each strain was recorded after 3 mM H_2_O_2_ was supplied into the growth medium. One representative result of three independent experiments is shown. NCM3722 (WT); CY325 (Δ*gcvB*); CY1057 (Δ*oxyR*). (**B**) Survival rate of the three strains after a 10-minute challenge of 20 mM H_2_O_2_. The survival rate is shown as the average ± S.D. of the six replicates in two independent experiments. *** *p* < 0.001 using Student’s *t*-test.

**Figure 6 antioxidants-10-01774-f006:**
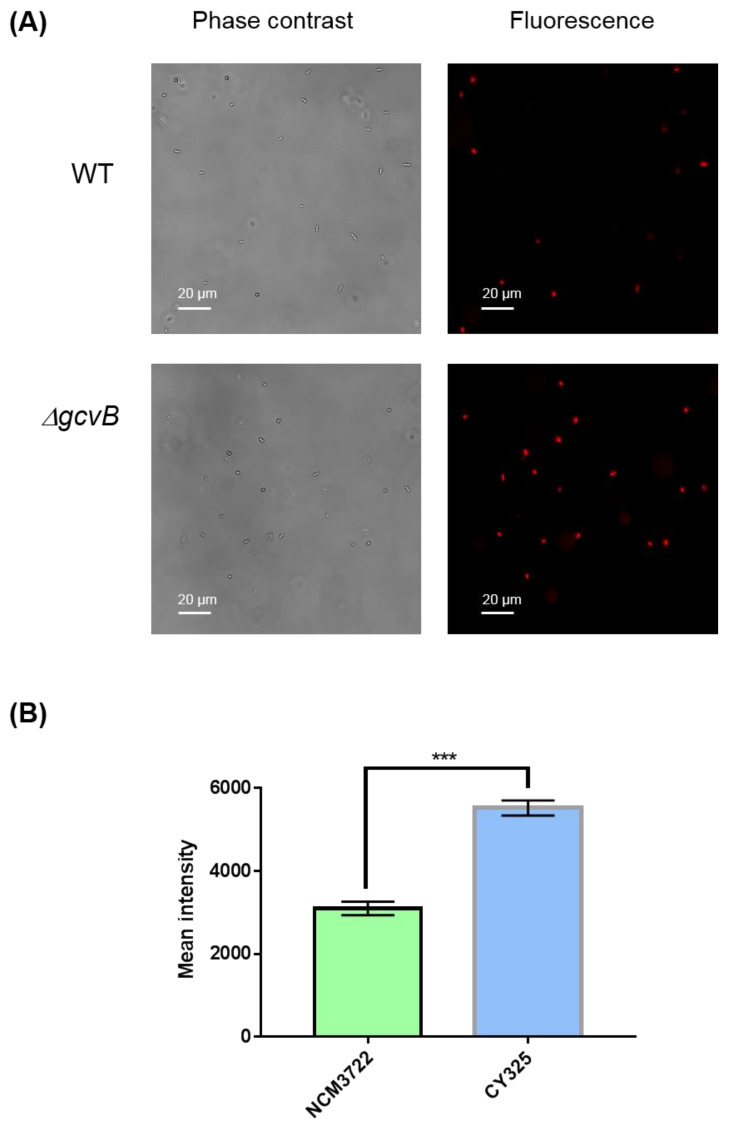
Detection of superoxide. (**A**) Detection of superoxide by DHE staining. The phase contrast shows the cells under microscope and the red fluorescence represents the detection of superoxide. Two to three independent observations were performed and one image containing representative cells is shown. (**B**) Quantification of fluorescence intensity. Fluorescence intensity was expressed as the average ± S.E.M., calculated from 393 to 416 individual cells. NCM3722 (WT); CY325 (Δ*gcvB*). *** *p* < 0.001 using Student’s *t*-test.

**Figure 7 antioxidants-10-01774-f007:**
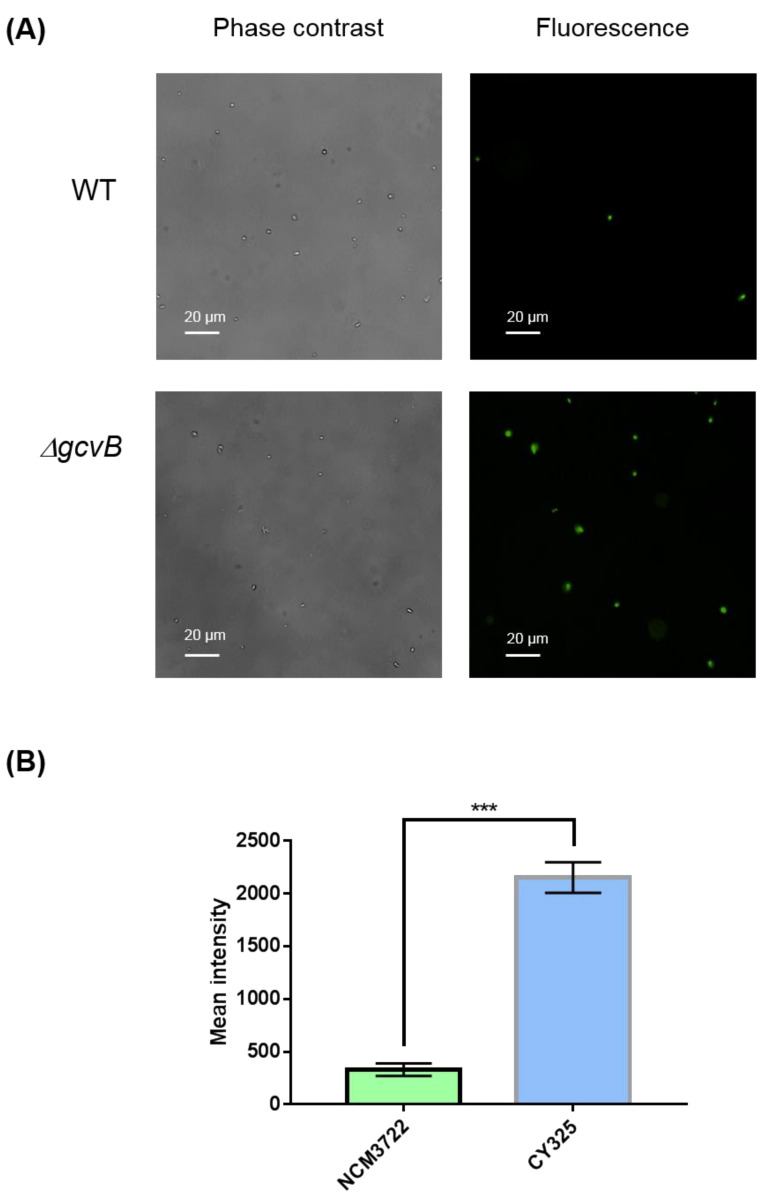
Detection of peroxide. (**A**) Detection of peroxide by H_2_DCFDA staining. The phase contrast shows the cells under microscope and the green fluorescence represents the detection of peroxide. Two to three independent observations were performed and one image containing representative cells is shown. (**B**) Quantification of fluorescence intensity. Fluorescence intensity was expressed as the average ± S.E.M., calculated from 468 to 741 individual cells. NCM3722 (WT); CY325 (Δ*gcvB*). *** *p* < 0.001 using Student’s *t*-test.

## Data Availability

The whole dataset of RNA-seq has been deposited to GEO with the accession number of GSE182531. All of the data is contained within the article and the Appendix A.

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
