# Peer review of "Small RNA GcvB Regulates Oxidative Stress Response of Escherichia coli"

_antioxidants, 2021, doi:10.3390/antiox10111774_

Round 1
Reviewer 1 Report
In this study, the authors performed comparative transcriptomes combined with molecular and physiological studies of the GcvB knockout and wild-type Escherichia coli strains grown in glucose minimal medium and identified putative some targets responding to the small RNA GcvB. The result revealed that the GcvB could regulate oxidative stress and enhances the E. coli’s survival ability by positively affecting the expression of OxyR at the translational level. The results seem interesting and valuable, but there are still some issues that need to be considered and solved to improve the manuscript.
- Figure 1A. Generally, the transcriptomes will have a huge data, here, only 11 points are seen in the Volcano map. Please confirm this is correct?
- Figure 2. A scale bar should be given in the microscope image.
- Figure 3A and 3B. If possible, please add the protein markers in the SDS-PAGE gel.
- In main text, authors mentioned: OxyR is the transcriptional regulator that induces the expression of antioxidant genes in response to oxidative stress. The enhanced ROS levels in the gcvB knockout strain could make this strain more sensitive to antibiotics treatment and GcvB could accordingly work as a putative antimicrobial target. But, here, only showing that gcvB knockout strain is more sensitive to oxidative stress, and did not verify the survival or death state of the E. coli. Could you give some data to see the survival and death of E. coli when in a high oxidative stress, like flow cytometry or other ways?
Reviewer 2 Report
The manuscript compared wild-type Escherichia coli K-12 strain transcripts and its gcvB deletion derivative grown in a minimal medium. Identified several novel targets putatively responding to GcvB, including flu, a determinant gene of auto-aggregation. In addition, molecular studies add layers to the oxidative stress regulator (OxyR) regulation.
The manuscript is attractive, well organized, with a clear description of objectives, well-presented results, and up-to-date references. Therefore, the manuscript is suitable for publication.
Minor points
The Escherichia coli bacteria should be revised and standardized throughout the text to italic.
